# Moderate Aerobic Exercise Induces Homeostatic IgA Generation in Senile Mice

**DOI:** 10.3390/ijms25158200

**Published:** 2024-07-27

**Authors:** Angel J. Hernández-Urbán, Maria-Elisa Drago-Serrano, Aldo A. Reséndiz-Albor, José A. Sierra-Ramírez, Fabiola Guzmán-Mejía, Rigoberto Oros-Pantoja, Marycarmen Godínez-Victoria

**Affiliations:** 1Laboratorio de Citometría de Flujo, Sección de Estudios de Posgrado e Investigación, Escuela Superior de Medicina, Instituto Politécnico Nacional, Mexico City 11340, Mexico; docjoelhernandez@gmail.com; 2Departamento de Sistemas Biológicos, Universidad Autónoma Metropolitana, Unidad Xochimilco, Mexico City 04960, Mexico; mdrago@correo.xoc.uam.mx (M.-E.D.-S.); fguzman@correo.xoc.uam.mx (F.G.-M.); 3Laboratorio de Inmunidad de Mucosas, Sección de Estudios de Posgrado e Investigación, Escuela Superior de Medicina, Instituto Politécnico Nacional, Mexico City 11340, Mexico; aresendiza@ipn.mx; 4Sección de Estudios de Posgrado e Investigación, Escuela Superior de Medicina, Instituto Politécnico Nacional, Mexico City 11340, Mexico; jsierrar@ipn.mx; 5Laboratorio de Neuroinmunoendocrinología, Facultad de Medicina, Universidad Autónoma del Estado de México, Toluca 50180, Mexico; rigoberto.orosp@gmail.com

**Keywords:** aerobic moderate exercise, aging, intestinal IgA, T-independent cell pathway, lamina propria, epithelial cells

## Abstract

A T-cell-independent (TI) pathway activated by microbiota results in the generation of low-affinity homeostatic IgA with a critical role in intestinal homeostasis. Moderate aerobic exercise (MAE) provides a beneficial impact on intestinal immunity, but the action of MAE on TI-IgA generation under senescence conditions is unknown. This study aimed to determine the effects of long-term MAE on TI-IgA production in young (3 month old) BALB/c mice exercised until adulthood (6 months) or aging (24 months). Lamina propria (LP) from the small intestine was obtained to determine B cell and plasma cell sub-populations by flow cytometry and molecular factors related to class switch recombination [Thymic Stromal Lymphopoietin (TSLP), A Proliferation-Inducing Ligand (APRIL), B Cell Activating Factor (BAFF), inducible nitric oxide synthase (iNOS), and retinal dehydrogenase (RDH)] and the synthesis of IgA [α-chain, interleukin (IL)-6, IL-21, and Growth Factor-β (TGF-β)]; and epithelial cells evaluated IgA transitosis [polymeric immunoglobulin receptor (pIgR), tumor necrosis factor-α (TNF-α), interferon-γ (IFN-γ), IL-4] by the RT-qPCR technique. The results were compared with data obtained from sedentary age-matched mice. Statistical analysis was computed with ANOVA, and *p* < 0.05 was considered to be a statistically significant difference. Under senescence conditions, MAE promoted the B cell and IgA+ B cells and APRIL, which may improve the intestinal response and ameliorate the inflammatory environment associated presumably with the downmodulation of pro-inflammatory mediators involved in the upmodulation of pIgR expression. Data suggested that MAE improved IgA and downmodulate the cytokine pro-inflammatory expression favoring homeostatic conditions in aging.

## 1. Introduction

Immunoglobulin A (IgA) is a key player in mucosal immunity by participating in protection against pathogens that colonize and/or invade the luminal surface and by collaborating in intestinal homeostasis [1]. The synthesis of IgA entails both T-cell-dependent (TD) and T-cell-independent (TI) pathways [2]. Intestinal IgA synthesis via T-cell-dependent and T-cell-independent pathways are under the control of intestinal microbiota [3,4] While most colonic commensals induce T-independent IgA generation, atypical commensals, such as filamentous bacteria and *Mucispirilum*, induce T-dependent IgA generation [3,4,5]. 

T-cell-independent pathway drives the constitutive IgA synthesis stimulated by luminal bacteria rendering low-affinity IgA and takes place in isolated lymphoid follicles (ILFs), lamina propria, and Peyer’s patches (PP) devoid of germinal center [6]. M cells transport luminal bacteria in the ILFs through a mechanism of transcytosis, and then dendritic cells (DCs) CD103+ recognize the luminal bacterial antigens to be delivered to B cells via the B cell receptor (BCR) in the epithelial dome. This recognition induces signal pathways in naïve B cells (IgM+ B cells), leading to the class switch recombination (CSR) of IgM+ B cells to IgA+ B cells under the stimulation of regulatory molecules released by DC CD103+, including B cell activating factor (BAFF) and the A Proliferation-Inducing Ligand (APRIL), and by epithelial cells (ECs), such as tumor necrosis factor-α (TNF-α), transforming growth factor-β1 (TGF-β1), and Thymic Stromal Lymphopoietin (TSLP). Additionally, T-cell-independent CSR factors include retinoic acid (RA) and nitric oxide (NO). Retinoic acid is obtained from diet, biliary acids, or/and endogenously by retinal dehydrogenase (RDH) expressed in DC CD103+ stimulated with flagellin via the toll like receptor (TLR)-5 signal. Nitric oxide is generated by inducible nitric oxide synthase (iNOS) [7].

In lamina propria, T-cell-independent IgA generation is led by cytokines released from DCs, such as TNF-α-induced NO production. Nitric oxide enhances the expression of TGF-β1 receptors on B cells to promote IgA synthesis. In addition, EC stimulates the recruitment of B cells to lamina propria through an expression of CCL20 and CCL28. Also, epithelial cells recognize luminal antigens through TLRs, promoting the production of TSLP, another CSR stimulatory factor that favors APRIL synthesis from lamina propria DCs to be targeted to B cells for IgA synthesis [8]. 

In addition, other transcription factors are involved in B cell development and differentiation impacting IgA production, such as Core-Binding Factor Alpha 3 (CBFa3/RUNX-3), Upstream Stimulatory Factor (USF), and paired box5 (Pax5). Furthermore, Pax5 is also critical for activating specific genes in pro-B cells or repressing genes in mature B lymphocytes and terminally differentiated IgA plasma cells, as well as IgA class switching [9]. 

Polymeric IgA is released by IgA+ plasma cells in the lamina propria and eventually interacts with the polymeric immunoglobulin receptor (pIgR) expressed at the basolateral membrane of the epithelial cell monolayer [10]. Both polymeric IgA and pIgR form a protein complex that is transported within endosomes to the apical surface where IgA bound to the protein pIgR derived fragment known as the secretory component (SC) is released in the lumen milieu as secretory IgA (SIgA) [4]. Experimental evidence shows that pro-inflammatory cytokines such as interleukin (IL-4), interferon-γ (IFN-γ), and TNF-α upregulate pIgR expression in intestinal epithelial cells, increasing the secretion of IgA [11,12].

The modulation of IgA synthesis by intestinal microbiota is under the control of exercise [13] and aging [14]. In humans, aging decrease taxon-specific IgA *Bifidobacteriaceae* and increased pathobionts, such as *Enterobacteriaceae* [14]. In mice, aging increased potential pathobionts (*Desulfovibrio* and *Bilophila*) and decreased IgA-associated and health-promoting bacteria (*Akkermansia* spp. *Lactobacillus* spp. and *Bifidobacterium* spp.) [15]. Contrarily, exercise induces the abundance of microbiota members, like *bifidobacteria* and *lactobacilli*, associated with IgA synthesis [16].

Aging of the immune system, or immunosenescence, leads to systemic deterioration with changes in the level of immunoglobulins and cytokines, which decreases the ability to react appropriately against pathogens, making us more susceptible to infections and other health problems [17,18]. Aging has subtle and complex effects on SIgA levels; average SIgA levels tend to increase with age until 60 years and then decrease slightly [19].

Scarce evidence of aging on intestinal IgA production has shown that total IgA concentration in intraluminal secretions of the small intestine from old rats is similar to adult ones [20]. Furthermore, aging increases monomeric IgA without changes in dimeric IgA in the intestinal lumen, and the binding capacity of dimeric IgA to bacteria is also reduced, which could be reflected in the quality and functionality of intestinal IgA in aging [21]. In addition, a lower specific IgA response in the jejunum and ileum in elderly rats immunized with choleric toxin has been identified [22]. Another study in mice immunized with cholera holotoxin also showed a higher number of IgA-secreting cells in lamina propria in both immunized and non-immunized old mice in comparison to young mice; however, serum-specific IgA concentration was lower in old rats [23]. Although experimental data support the effect of aging on IgA production by the T-dependent pathway, there is no evidence of aging-induced changes in IgA production by the TI pathway. 

The effect of aging on the transcytosis of IgA in the intestinal epithelium shows that a low level of urine IgA in ddYHigh mice is caused by a deficiency of transcytosis of IgA in the intestinal mucosa. Therefore, the age-dependent changes in the pIgR expression but not in the IgA structure seem to be a possible cause for the elevation of serum IgA [24]. 

At present, exercise has been shown to provide protective outcomes to immune dysfunctions associated with senescence [25,26,27]. As reported previously by our research group, moderate exercise enhanced the IgA antibody response, as found in the small intestine of mice [28,29]. In other studies, moderate exercise plus a final bout of strenuous exercise promotes IgA increase and also downmodulates some mediators involved in its synthesis and transport in young adult mice (6 months old) [30]. It is thought that exercise modulates the IgA response by proving stress activation of the hypothalamus–hypophysis–adrenal axis, resulting in the release of corticosteroid hormones, like cortisol. In previous contributions, moderate exercise displayed differential outcomes either by enhancing cortisol levels in adult young mice or reducing them in elderly mice (data previously published by Sierra-Ramírez et al., 2022) [31].

## 2. Results

Data were analyzed by comparing adult (6 months) and senescent mice (24 months) versus the baseline results from young mice (3 months). 

The experimental strategy of flow cytometry analysis of the B cell and plasma cell sub-populations is shown in Appendix A, respectively. 

### 2.1. Exercise Increased Total B Cells in Senile Mice

Comparisons among the sedentary groups evidenced that total B cell percentage (%) was decreased in the elderly compared to young mice and adult mice (both *p* < 0.001, see Figure 1a). Concerning their respective age-matched sedentary group, B cell % was either lower in adult exercised mice or higher in the senile exercised group (adult *p* < 0.05, elderly *p* < 0.001, Figure 1a). Within the sedentary groups, IgM+ B cell % was lower in adults and the elderly than in young mice (adult *p* < 0.001, elderly *p* < 0.05). Within the exercised groups, IgM+ B cell % was greater in elderly versus adult mice (*p* < 0.05, Figure 1b).

### 2.2. Exercise Increased the IgA+ B Cells in Senile Mice

Analysis among the sedentary mice indicated that IgA+ B cell % was found to be greater in adult and elderly mice than young mice (adult *p* < 0.001, elderly *p* < 0.05, Figure 1c) and lower in elderly mice than adult mice (*p* < 0.01, Figure 1c). Comparisons within the age-matched groups evidenced that IgA+ B cell % was greater in the senile exercised group than in the senile sedentary group (*p* < 0.05, Figure 1c).

Analysis within the sedentary groups indicated that plasma cell % was lower in adults than young mice or greater in senile than adult mice (both *p* < 0.01, Figure 1d). Within the age-matched groups, plasma cell % was greater in adults exercised than in adult sedentary mice (*p* < 0.001, Figure 1d).

Comparative analysis with the sedentary groups showed that IgM+ plasma cell % was lower in adult versus young mice or higher in senile versus adult mice (both *p* < 0.001, Figure 1e). Analysis within the age-matched groups indicated that IgM+ plasma cell % was higher in adult exercised mice versus adult sedentary mice or lower in exercised elderly mice than aged sedentary mice (both *p* < 0.001, Figure 1e)

Analysis within the sedentary mice evidenced that IgA+ plasma cell % was lower in adult mice than young mice or higher in senile mice than adult mice (adult *p* < 0.05, senile *p* < 0.001, Figure 1f). Within the age-matched groups, IgA+ plasma cell % was greater in adult exercised mice than in adult sedentary mice (*p* < 0.01, Figure 1f). 

### 2.3. Exercise Increased the α-Chain and APRIL mRNA in Adult and Senile Mice

Within aged-paired groups, α-chain mRNA was found to be greater in the adult and senile exercised mice than in their respective sedentary control groups (adult *p* < 0.001, senile *p* < 0.01, Figure 2a). Moreover, the α-chain mRNA level was greater in senile exercised mice than adult exercised mice (*p* < 0.01, Figure 2a).

Comparative analysis of TSLP mRNA evidenced a greater level in exercised mice than sedentary adult mice and in sedentary aged mice versus sedentary adult mice (both *p* < 0.05); a lower TSLP mRNA level was found in senile exercised mice than adult exercised mice (*p* < 0.001, Figure 2b).

Analysis of APRIL mRNA evidenced that it was greater in adult and senile exercised mice versus their respective sedentary control groups (both *p* < 0.001, Figure 2c).

Comparative analysis of BAFF and RDH mRNA within the adult mice evidenced greater levels in exercised mice than in sedentary mice (BAFF *p* < 0.01, Figure 2d; RDH *p* < 0.05, Figure 2e). Within the sedentary groups, the iNOS mRNA level was found to be greater in elderly mice versus young and adult mice (both *p* < 0.05, Figure 2f). Within the exercised groups, TGF-β1 mRNA was seen to be higher in elderly mice than adult mice (*p* < 0.01, Figure 2g). Regarding IL-6 mRNA, no significant differences were seen (Figure 2h).

### 2.4. Factors Related to B Cells Maturation and Differentiation 

Within age-paired groups, CBFa3 mRNA levels were significantly increased in senile exercised mice versus senile sedentary mice (*p* < 0.001, Figure 3a). In the USF-1 mRNA level, no significant differences were seen at 6 or 24 months (Figure 3b). However, Pax5 mRNA levels were significantly decreased in senile exercise mice versus sedentary senile mice (*p* < 0.001, Figure 3c). Regarding aging, in terms of CBFa3, Pax5, and USF mRNA levels, no significant differences were seen in senile sedentary mice versus young mice (Figure 3).

### 2.5. Exercise Suppressed Parameters of pIgR for IgA-Transcytosis

Within aged-paired groups, the pIgR mRNA level was found to be either greater in the adult exercised group versus the adult sedentary group or lower in the senile exercised group versus the senile sedentary group (both *p* < 0.001, Figure 4a). Within the exercised groups, the pIgR mRNA level was significantly lower in senile mice versus adult mice *p* < 0.001, Figure 4a).

Comparisons within the sedentary groups indicated that the IL-4 mRNA level was seen to be higher in elderly mice versus young and adult mice (both *p* < 0.01, Figure 4b). Moreover, within the senile mice, IL-4 mRNA was seen to be lower in the exercised group than sedentary group (*p* < 0.01, Figure 4b).

Within the sedentary mice, the IFN-γ mRNA level was found to be higher in elderly mice versus young mice (*p* < 0.001, Figure 4c). Within the age-matched groups, IFN-γ mRNA levels were seen to be lower in the exercised groups than the sedentary groups (all *p* < 0.001, Figure 4c). Regarding TNF-α RNA, no significant differences were seen (Figure 4d). 

## 3. Discussion

It is well recognized that physical activity has a substantial impact on components of intestinal immunity, microbiota, and IgA production in aging [32,33]. At present, the evidence about the T-cell pathways involved in the production of IgA in the small intestine of elderly mice is minimal. The outcome of MAE in elderly mice in terms of the T-dependent pathway was reported before [34]. Thus, in the current assay conducted in the same experimental mice groups, the impact of MAE was analyzed, for the first time, in terms of a T-independent pathway.

Comparative analysis between exercised versus sedentary age-matched groups indicated that MAE upmodulated α-chain mRNA in both senile and adult mice, although it favored the response of immature cells (B cells and IgA+ B cells) in senile mice and the response of mature cells (including plasma cells, IgM+, and IgA+ plasma cells) in adult mice. It is known that aging downmodulates the B cell populations but paradoxically increases the antibody production attributable to dysregulation of the mucosal immune system [35]. Thus, data suggested that MAE drives divergent B cell responses in aged and adult mice according to B cell maturity. A recent contribution from our group of study [34] indicated that MAE increased intestinal IgA antibody concentration via T-cell dependence in adult mice only. In senile mice, MAE induced an increase in homeostatic IgA, presumably via microbiota modulation [34]. We assume that homeostatic IgA (i.e., “innate” IgA) was generated via a T-independent pathway evaluated in lamina propria and depleted of Peyer´s patches as a major compartment where the T-dependent IgA pathway takes place. In the current assay, interestingly, MAE only enhanced APRIL mRNA levels in senile mice, whereas it elicited TSLP, APRIL, BAFF, and RDH mRNA levels in adult mice. Data suggested that MAE enhanced the APRIL mRNA level associated with the T-cell-independent pathway of the intestinal immune response of IgA. The last findings are very important in aging because it has been described that homeostatic IgA can eliminate some pathogens, such as rotavirus and *Salmonella typhimurium,* and thereafter the pathogen-specific high-affinity IgA is secreted within the intestinal lumen [36]. As claimed by others, the T-cell-independent pathway temporarily replaces the T-cell-dependent pathway during the early phase of infection [2]. The impact of MAE on the upmodulation of a specific IgA antibody response with a protective outcome against infection was reported in the model of *Salmonella typhimurium* infection in 29-week-old BALB/c mice [36]. Human trials and experimental settings indicate that aging blunts the CSR that drives the B cells committed to antibody production and immunologic memory, as described in the systemic compartment [37]. These findings suggested that MAE improved intestinal IgA concentration by enhancing the APRIL mRNA level and IgA+ B cell % in LP associated with the T-cell-independent pathway. The findings support the impact of MAE on the increase (twofold) of intestinal IgA concentration in aged mice by comparison with senile sedentary mice [34]. 

On the other hand, among transcript factors related to efficient IgA class switching is RUNX3, which is an important downstream target of TGF-β signaling to activate Sa germline transcription (GLT) [38]. The results of this study indicate that MAE promotes RUNX3 mRNA expression in adulthood and aging, contributing to IgA CSR. 

Cell intrinsic mechanisms of survival of mature plasma cells entail the downregulation of Bcl6 and Pax5, as well as the upregulation of the transcription factors IRF4 and Blimp-1 [39]. In addition, the loss of Pax5 levels during terminal differentiation contributes to the plasma cell transcription program to IgA CSR [9]. Despite the plasma cell number being unchanged by MAE, in this study, increased luminal IgA concentration (previously reported [34]) and the α-chain mRNA level found in exercised mice may result from the decrease in Pax5 mRNA, favoring both IgA plasma cell survival and IgA CSR.

In this study, MAE increased the pIgR mRNA level in adult mice, as also documented in 8-week-old (young) BALB/c mice [29]. In the current assay, IL-4 and TNF-α were unaffected, while IFN-γ mRNA was decreased by MAE in adult mice. Previous studies documented that IL-4 and TNF-α levels were upmodulated by MAE in the duodenum of the small intestine in young BALB/c mice [29]. Apparent discordances may reflect the impact of age on cytokine-mediated pIgR expression. Although pIgR expression is constitutive, it can be regulated by IL-4 and pro-inflammatory cytokines, such as TNF-α and IFN-γ [40]. 

Furthermore, MAE decreased pIgR mRNA in aged mice along with IL-4 and IFN-γ mRNA. pIgR mRNA expression is controversial given that it is unaffected by aging, as documented in senile male Fisher rats [41], but it was affected by diet in six-week-old male C57BL/6 mice [42]. Studies in the differentiated intestinal epithelial cell have shown that a basal expression of pIgR is enhanced by USF proteins [43], which enhanced the binding promoter activity of the pIgR gene; in this study, changes in USF1 mRNA levels were not seen in exercised mice, suggesting that USF1 activity was not related with the decrease in pIgR expression. The impact of aging on pIgR expression has been evidenced in dYY mice (with high monomeric IgA concentration in serum and low monomeric IgA concentration in urine) and is associated with defective pIgR-mediated IgA transport [24]. In a previous study, we reported that luminal IgA was increased in senile exercise mice [34]; in the current study, pIgR mRNA expression was found to decrease, however role of pIgR in IgA transport under senescence and MAE conditions must be clarify considering that pIgR protein expression was not assayed. Pro-inflammatory cytokines, including IFN-γ, TNF-α, and IL-4, favor pIgR mRNA expression; thus, data suggested that MAE under senescence conditions downmodulates the pro-inflammatory cytokines and upmodulates the anti-inflammatory cytokines, such as IL-10 and TGF-β. Although, in the current contribution, IL-6 mRNA expression was unchanged in exercised elderly mice at lamina propria, in previous contributions, moderate exercise increased IL-6 mRNA expression (but not TNF-α and IFN-γ) in the skeletal muscle during young and aging stages [44]. Furthermore, exercise-induced IL-6 has anti-inflammatory effects, favoring the secretion of IL-10, a soluble TNF receptor, and the antagonist of the IL-1 receptor [44]. In this study, MAE in elderly mice induced an unsignificant decrease in TNF-α along with a significant decrease in IFN-γ, suggesting a potential anti-inflammatory outcome in lamina propria in the small intestine.

Therefore, MAE may provide a beneficial impact on aging by both ameliorating the cytokines that participate in the inflammatory process and by increasing the luminal IgA. Thus, MAE contributes to the regulation of the immune system at the mucosal level and ultimately in the intestinal homeostasis in aging.

A comparative analysis between sedentary groups, both adult and elderly mice, indicated that aging decreased B cell % and increased IgA+ B cell % versus young mice, whereas IgM+ B cells and total, IgM+, and IgA+ plasma cells were upmodulated in the adult group. These findings agreed with the impact of aging on the upmodulation of lamina propria plasma cells and the decrease in circulant IgA in senile rats [23,35]. In other assays, a decrease in surface IgA (sIgA) cells and an increase in sIgM cells were documented in PP in senile mice [45]. Apparent divergences may be derived from the inductive versus effector site of analysis: PP in the previous study [45] and lamina propria in the current. Furthermore, these discordances may reflect the effect of senescence on the reduction of B cell precursors of IgA-secreting plasma cells and PP function and cellularity [23]. Moreover, age impacts the development and function of ILFs as potential sites for T-independent IgA intestinal response by decreasing the B cell population and increasing T cells [35]. In this study, decreased B cell population in lamina propria may result from decreased mRNA and protein expression in CCL20 and CXCL13 chemokines involved in B cell recruitment into ILFs, as documented previously [35]. In this study, aging increased iNOS and IL-4 mRNA versus young and adult mice and increased IFN-γ mRNA versus young mice only, while no changes in pIgR mRNA were seen. Data suggested that aging promoted the pro-inflammatory cytokine levels associated with the increase in pIgR (IL-4 and IFN-γ) [11,12,14], and presumably, these cytokines could have maintained pIgR expression in the aged mice to the same extent as young and adult mice. Finally, aging increased iNOS mRNA levels versus young and adult mice. It is known that iNOS is a factor associated with the T-cell-independent pathway. 

Although IgM and IgA protein expression were evaluated by flow cytometry, one of the most important limitations of this study is that factors associated with class switch recombination of IgM → IgA and those associated with the expression of pIgR were only evaluated at the transcriptional level. In addition, at the luminal level of IgA bacteria complexes was not assessed to address potential relationships between the mucosal parameters and microbiota with the critical role on IgA responses via T-independent pathways. Despite these limitations, we can conclude that MAE improved the T-independent IgA response in small intestine. In the elderly, MAE seems to provide an anti-inflammatory environment by decreasing the pro-inflammatory markers (i.e., IL-4 and IFN-γ) that are involved in the increase in pIgR expression.

## 4. Materials and Methods

### 4.1. Animals

Six-week-old male BALB/c mice (N = 40) were obtained from “Unidad de Producción y Experimentación de Animales de Laboratorio, Universidad Autónoma Metropolitana Unidad Xochimilco (UPEAL UAMX)”. The mice were housed in groups of five per cage with light–dark cycles of 12:12 h (7 AM/7 PM) at a relative humidity of 55% and at a temperature of 20 °C ± 5 °C. The mice were provided with water and fed with Laboratory Rodent Diet 5001 (LabDiet, Saint Louis MO, USA) ad libitum. Manipulation and the exercise protocol were conducted between 8 AM and 11 AM to avoid the influence of circadian cycles of adrenocorticotropic hormone (ACTH) and corticosterone [46]. The procedures carried out on the animals complied with the requirements that determine the NOM-062-ZOO-1999 “Technical specifications for the production, care, and use of laboratory animals” SAGARPA standard and the “Guide for the Care and Use of Laboratory Animals”. This protocol was approved by the Institutional Committee for the Care and Use of Laboratory Animals of the Higher School of Medicine of the National Polytechnic Institute, Mexico, with register key ESMCICUAL_04/08-11-2020. 

### 4.2. Experimental Design

The animals were adapted to their environment for 2 weeks before starting the exercise protocol on week 9 of life. The mice were divided into 5 groups: (1) toung sedentary group, sacrificed at 3 months of age, (2) sedentary adult group, sacrificed at 6 months of age, (3) exercised adult group, exposed to the MAE protocol at up 6 months of age, (4) senile sedentary group, sacrificed at 24 months of age, and (5) exercised senile group, exposed to the MAE protocol at up to 24 months old. Groups of sedentary mice experimentally un-manipulated and fed with a standard diet were included as controls.

### 4.3. Moderate Aerobic Exercise Protocol

Before starting the exercise protocol, the mice underwent an adaptation period during weeks 9 and 10 of life. The adaptation period consisted of a 10 min/day session of exercise on the endless band at a speed of 5 m/h and 0° of inclination with increments of 5 min and 5 m/h every other day until reaching 30 min/day at a speed of 25-30 m/h. After that, the mice ran on an endless band at a speed of 30 m/h at 0° of inclination for 30 min/day, 5 days/week, and rested for 2 days (on weekends) until 6 or 24 months old [31].

### 4.4. Sampling

At 6 or 24 months of age, sedentary and exercise mice were sacrificed by administering a lethal dose of sodium pentobarbital at 150 mg/kg body weight by intraperitoneal via. After that, the small intestine was dissected, and intestinal lavage with PBS was performed. After intestinal lavage, Peyer patches were discarded, and epithelial cells and lamina propria were obtained using different Percoll gradients, as described previously by Resendiz-Albor et al., 2010 [47]. The cells obtained were separated and analyzed by flow cytometry or stored at −70 °C for posterior analysis by RT-qPCR. 

### 4.5. Lymphoid Sub-Populations in Lamina Propria by Cytometry 

The percentage of the following cell populations was determined: naïve B cells [IgM+/IgD+/B220+/CD19+], membrane IgA+ B cells [IgA+/B220+/CD19+], and cytoplasmic IgA plasma cells [IgA+/CD138+] in the lamina propria of the small intestine using the flow cytometry technique. The staining of the lymphocytes was carried out using the next anti-mouse antibodies from Becton Dickinson (BD) Technologies: PECy7-CD19 (cat. 561739, BD Pharmigen, Franklin Lakes, NJ, USA), PE-CD45R/B220 (cat. 553089, BD Pharmigen), APC-IgD (Cat. 560868, BD Pharmigen), PerCP-Cy5.5-IgM (cat. 550881, BD Pharmigen), and FITC-IgA (cat. 559354, BD Pharmigen) obtained from Pharmingen (San Diego, CA, USA). The protocol provided by the manufacturer was used for surface staining. After staining plasma cells from LP, cells were permeabilized and fixed with a cytoperm/cytofix kit (cat. 554722 Becton Dickinson, Franklin Lakes, NJ, USA) for cytoplasmic IgA determination. The samples were acquired in a BD FACSAria Fusion (BD 400, BD Biosciences, San Jose, CA, USA) flow cytometer using FACSDiva v.8.0.2 acquisition and analysis software (BD Biosciences, San Jose, CA, USA). Ten thousand events in the lymphocyte region in the SSC vs. FSC dot plot for B cells and 20,000 events for plasma cells were analyzed. Data were analyzed in FlowJo^TM^ Software version 7.6.2 for Windows, Becton Dickinson (Ashland, OR, USA). 

### 4.6. Relative Expression of mRNA Assay

The relative mRNA levels of genes related to IgA-synthesis (α-chain, TSLP, APRIL, BAFF, iNOS, RDH, TGF-β1, and IL-6 mRNA) and B cell differentiation and maturation (CBFa3/RUNX-3, USF-1, and Pax5) were determined in the cell suspensions of lamina propria. mRNA levels of IL-4, TNF-α, IFN-γ, and pIgR were assessed in the suspensions of epithelial cells by RT-qPCR.

Total RNA extraction was performed using the protocol described for the fabricant of the TRIzol^®^ reagent (cat. 15596018, Invitrogen, Carlsbad, CA, USA). The concentration and purity of total RNA were determined by spectrophotometry in the Nanodrop™ 2000/2000c (Cat: ND-2000, Thermo Scientific, Wilmington, DE, USA). The ratio of absorbances A260/280 with a value between 2.0-2.2 and > 1.7 was considered an optimal purity. RNA integrity was analyzed by electrophoresis in the gel of agarose at 1%. Reverse transcription was performed using the Maxima First cDNA Synthesis Kit for RT–qPCR (cat. No. K1642, Thermo Scientific, Vilnius, Lithuania) at 37 °C for 50 min. 

Quantitative PCR was subsequently performed by a duplicate using a Maxima Probe/ROX qPCR Master mix Kit (cat. K0261, Thermo Scientific™, Vilnius, Lithuania) in a LightCycler ^®^ Nano Instrument (Roche Diagnostics GmbH, Rotkreuz Switzerland) at an initial denaturing step (at 95 °C for 10 min) followed by 45 cycles of amplification (at 95 °C for 10 s, 60 °C for 35 s, and 72 °C for 1 s) and 1 cycle of cooling (40 °C for 30 s).

Specific oligonucleotide primers were originally generated by using the online assay design software (ProbeFinder: http://www.universal-probelibrary.com accessed on 1 September 2017.) and the primer sequences for TSLP, APRIL, BAFF, iNOS, RDH, TGF-β1, IL-6, CBFa3/RUNX-3, USF-1, Pax5 and, GAPDH, as shown in Table 1. A primer sequence of α-chain, pIgR, IFN-γ, IL-4, and TFN-α was previously published by Godinez-Victoria, et al., 2012 [30]. Messenger RNA expression levels were calculated using the comparative parameter quantification cycle method and normalized to the relative expression of GAPDH mRNA [48].

### 4.7. Statistical Analysis

At the end of this study (at 24 months old), the survival was ~60% in the control group and ~80% in exercised group. For that, data are presented as the mean and standard deviation (SD) of 6–8 animals per group. For the comparison between groups in the different measurement times, two-way ANOVA was used considering the group (sedentary or exercised) and the age of the mice (young, adult, and elderly mice) as the factors, followed by a Turkey method post hoc test. For all tests, *p* < 0.05 was considered significant. All data were analyzed using SigmaPlot for Windows version 11.1 (Systat Software Inc. San Jose CA, USA). *p* < 0.05 was considered to indicate a statistically significant difference.

## 5. Conclusions

This study evidenced that MAE has differential outcomes on factors associated with IgA generation according to the age of mice. MAE enhanced the T-cell-independent pathway in both adult and senile mice. Under senescence conditions, MAE promoted the B cell and IgA+ B cells and APRIL mRNA, which may improve the intestinal response. Under senescence conditions, MAE improved IgA response via the T-independent pathway by downmodulating the molecular markers that favored pIgR expression, i.e., IL4 and IFN-γ. Finally, this study may provide foundations of moderate exercise as a strategy to enhance immunity to protect against infectious diseases in elderly people.

## Figures and Tables

**Figure 1 ijms-25-08200-f001:**
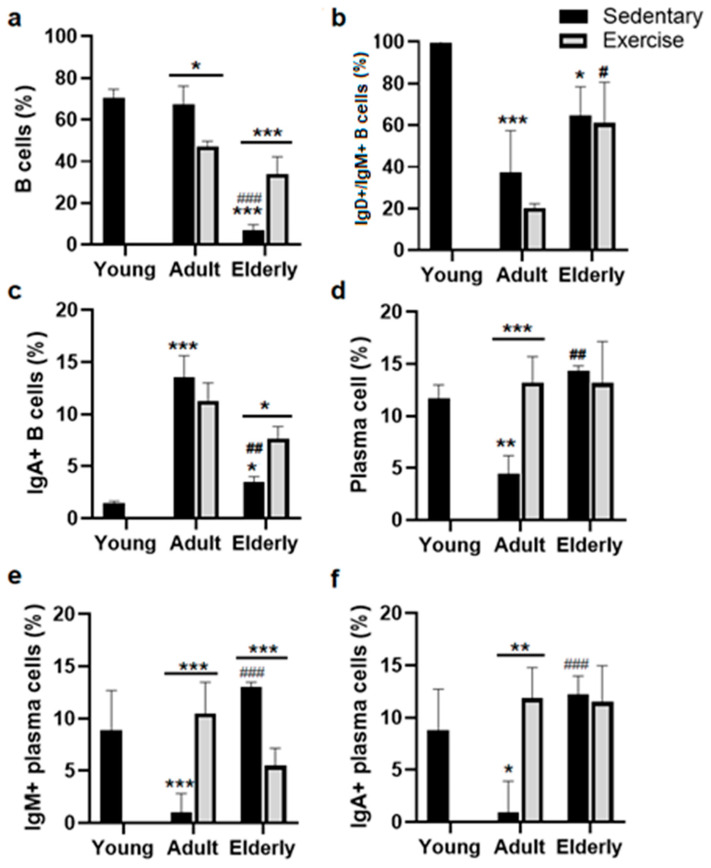
Lymphoid sub-populations in lamina propria in the small intestine of sedentary and exercised mice at different ages. Data represent the mean and standard deviation of (**a**) total B cells; (**b**) membrane IgM+/IgD+ B cells; (**c**) membrane IgA+ B cells; (**d**) plasma cells; (**e**) cytoplasmic IgM+ plasma cells; (**f**) cytoplasmic IgA+ plasma cells. (*) bar versus young mice; (#) bar versus respective adult control; asterisks upon line on the bars versus respective sedentary control. * *p* < 0.05, ** *p* < 0.01, and *** *p* < 0.001; # *p* < 0.05, ## *p* < 0.01, and ### *p* < 0.001.

**Figure 2 ijms-25-08200-f002:**
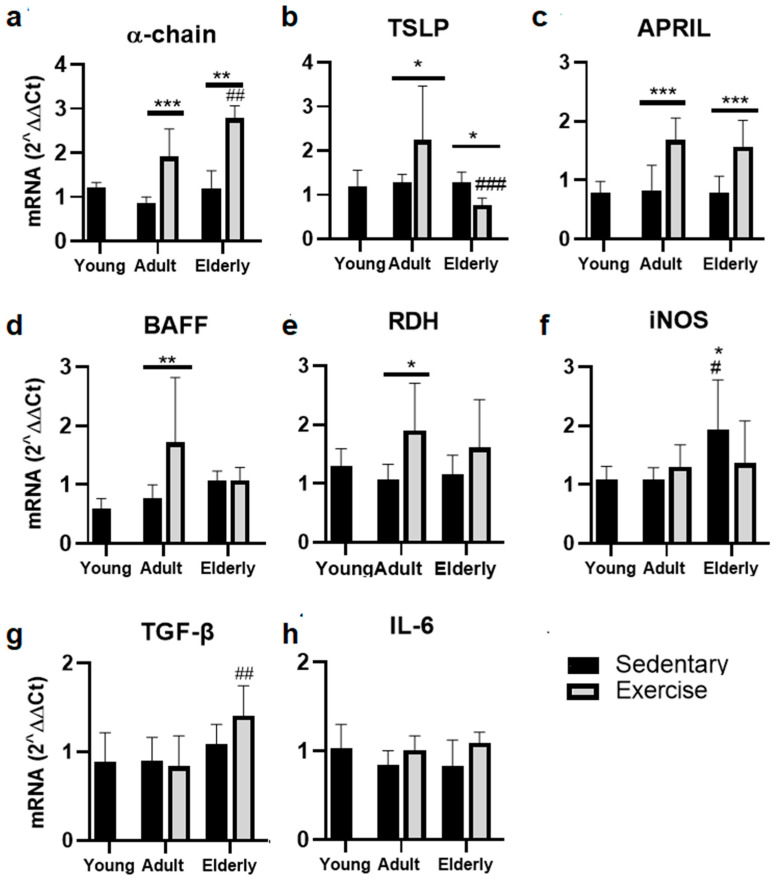
Gene expression of factors related to the production of IgA in lamina propria in the small intestine in sedentary or exercise mice at different ages. Data represent the mean and standard deviation of (**a**) α-chain, (**b**) Thymic Stromal Lymphopoietin (TSLP), (**c**) A Proliferation-Inducing Ligand (APRIL), (**d**) B cell activating factor (BAFF), (**e**) retinol dehydrogenase (RDH), (**f**) inducible nitric oxide synthase (iNOS), (**g**) transforming growth factor β (TGF-β), (**h**) interleukin (IL)-6. (*) bar versus respective young mice; (#) bar versus respective adult control; asterisks upon a line on the bars versus respective sedentary control. * *p* < 0.05, ** *p* < 0.01, and *** *p* < 0.001; # *p* < 0.05, ## *p* < 0.01, ### *p* < 0.001.

**Figure 3 ijms-25-08200-f003:**
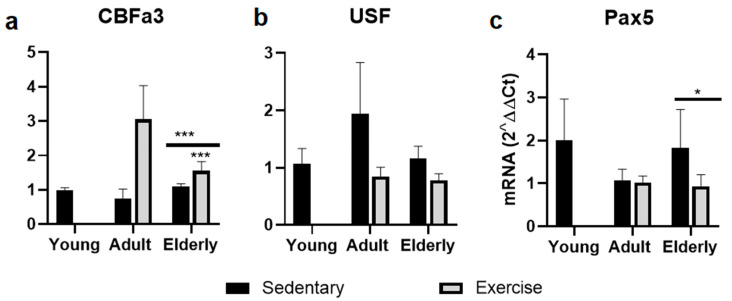
Gene expression of transcription factors involved in B cell development and differentiation impacting IgA production. Data represent the mean and standard deviation of (**a**) Core-Binding Factor Alpha 3 (CBFa3/RUNX-3), (**b**) Upstream Stimulatory Factor-1 (USF), and (**c**) paired box5 (Pax5). (*) bar versus young mice or asterisks upon line on the bars versus respective sedentary control. * *p* < 0.05 and *** *p* < 0.001.

**Figure 4 ijms-25-08200-f004:**
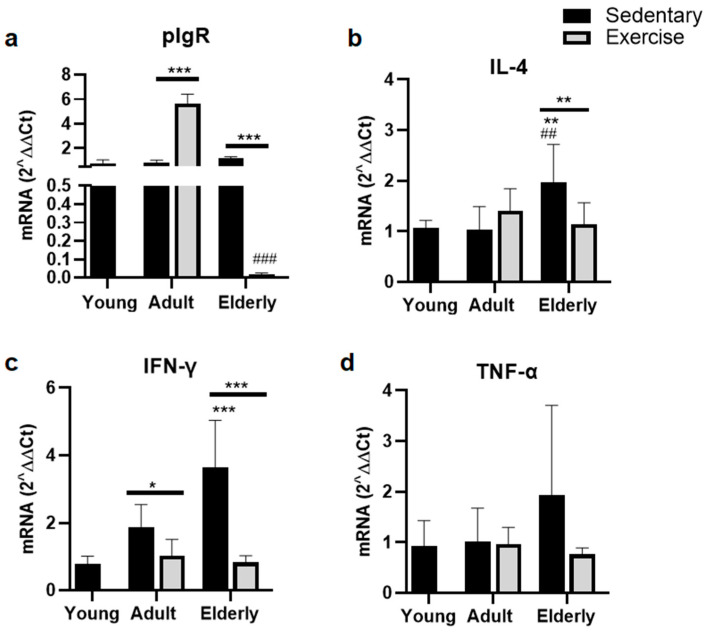
Gene expression of the polymeric immunoglobulin receptor in epithelial cells (**a**) and cytokines related with its expression in lamina propria (**b**,**c**) in the small intestine in sedentary mice or exercise mice at different ages. Data represent the mean and standard deviation of (**a**) pIgR, (**b**) interleukin (IL)-4, (**c**) interferon-γ (IFN)-γ, and (**d**) tumor necrosis factor-α (TNF)-α. (*) bar versus respective young mice; (#) bar versus respective adult control; asterisks upon line on the bars versus respective sedentary control. * *p* < 0.05, ** *p* < 0.01, and *** *p* < 0.001; ## *p* < 0.01 and ### *p* < 0.001.

**Table 1 ijms-25-08200-t001:** Forward and reverse primers for real-time PCR assays designed according to the ensemble transcript ID of the Universal ProbeLibrary.

Gen	Sequence 5′→3′		ID
	Forward	Reverse	
*TSLP*	cag ctt gtc tcc tga aaa tcg	aaa tgt ttt gtc ggg gag tg	NM_021367.2
*BAFF*	atg cgg aag gca gat tga	tgc atc ttt tgc tac cct ga	NM_001347309.1
*APRIL*	agc tgg gca ctg agc ttt ac	aag ttg gcc tcg aac tca tc	NM_023517.2
*iNOS*	ctt ttc cta tgg ggc aaa aa	ctg gaa ctc tgg gct gtc a	NM_010927.4
*RDH11*	tgt act tgg tca cgc caa aa	ccg gga agc tga aca tta ga	NM_021557.5
*TGF-β1*	tgg agc aac atg tgg aac tc	gtc agc agc cgg tta cca	NM_011577.2
*IL-6*	gct acc aaa ctg gat ata atc agg a	cca ggt agc tat ggt act cca gaa	NM_031168.2
*CBFa3*	gct ctc tca gca cca cga g	tca ggt ctg agg agc ctt g	NM_019732.2
*USF-1*	tca aga ggt ggg aaa gga tg	cat tgg gcc ccc ttc tac	NM_001305676.1
*Pax5*	cct ggg agt gaa ttt tct gg	tgg gga acc tcc aag aat c	NM_008782.2
*GAPDH*	aag agg gat gct gcc ctt ac	cca ttt tgt cta cgg gac ga	NM_001289726.1

## Data Availability

The original contributions presented in the study are included in the article, and further inquiries can be directed to the corresponding author.

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
