# Peer review of "Moderate Aerobic Exercise Induces Homeostatic IgA Generation in Senile Mice"

_ijms, 2024, doi:10.3390/ijms25158200_

Round 1

Reviewer 1 Report

Comments and Suggestions for Authors

Some of the experimental results of this paper cannot support the conclusion, and the quality of the manuscript needs to be further improved through additional data:

1. All the results in this paper are based on the conditions of non-T cell-dependent response. In Figure 1, exercise can induce changes in the number of B cells and the number of IgA-expressing B cells in aged mice, and even the number of IgM-expressing plasma cells. Significant changes have taken place, so it is necessary to trouble the author to provide the results of changes in T cell response in exercise group and sedentary group mice, and then to exclude the activation of T cell response during the experiment.

2. First of all, there is only the processed data in Figure 1, and could the author please provide the specific results of flow cytometry. Then exercise can increase the number of B cells expressing IgA in elderly mice, so whether sIgA in the intestine also increases? This result is very important to support the author’s conclusion.

3. In Figure 2, 3 and 4, there was only the change of transcription level, and there was no experimental data of protein level change. In addition, are the changes in the transcription levels of IF-4 and IFN-γ related to Th cell response? The author should provide relevant explanations and proofs.

4. The important conclusion of the author 's article is that exercise leads to a change in the number of B cells expressing IgA in the intestine of aged mice, which down-regulates the intestinal pro-inflammatory response. However, in Figure 2 and Figure 4, the important pro-inflammatory factors in small intestine cells, IL-6 and TNF-α transcription levels did not change significantly. In addition, there was no data on the changes of pro-inflammatory factor IL-1. The authors need to provide relevant explanations and supplements for the changes of these important pro-inflammatory factors.

5. The author puts forward the conclusion that exercise leads to changes in the expression of IgA B cells in the intestinal tract of elderly rats, thus down-regulating the intestinal pro-inflammatory response. Then, whether the changes of inflammatory factors (qPCR results) mentioned in this paper are related to the changes of IgA levels, the author’s data cannot fully support this view.

6. Does the sedentary mice mentioned in the author’s article do not do any treatment and are normally fed mice? If you can, please give a certain narrative in the article, do not cause ambiguity.

Author Response

Some of the experimental results of this paper cannot support the conclusion, and the quality of the manuscript needs to be further improved through additional data:

Comment 1: All the results in this paper are based on the conditions of non-T cell-dependent response. In Figure 1, exercise can induce changes in the number of B cells and the number of IgA-expressing B cells in aged mice, and even the number of IgM-expressing plasma cells. Significant changes have taken place, so it is necessary to trouble the author to provide the results of changes in T cell response in exercise group and sedentary group mice, and then to exclude the activation of T cell response during the experiment.

Response 1: Thank you very much for your comment. Previously, we reported that T cell-dependent immune response was not activated in senile mice under moderate aerobic exercise conditions as it was reported in Figure 2 by Hernández-Urbán AJ, et al., 2023. See please the figure below.

Due the T cell-dependent pathway takes place largely at Peyer´s patches, these structures were fully removed to evaluate the T cell-independent response only in lamina propria. Furthermore, innate (CD3-) T cells present in lamina propria, can stimulate the B-cells but these cells only induce the T cell-independent response.

  • Hernández-Urbán AJ, Drago-Serrano ME, Cruz-Baquero A, García-Hernández AL, Arciniega-Martínez IM, Pacheco-Yépez J, Guzmán-Mejía F, Godínez-Victoria M. Exercise improves intestinal IgA production by T-dependent cell pathway in adults but not in aged mice. Front Endocrinol (Lausanne). 2023 Dec 7;14:1190547. doi: 10.3389/fendo.2023.1190547.

 Comment 2: First, there is only the processed data in Figure 1, and could the author please provide the specific results of flow cytometry.

Response 2: Thank you. Dot-plots of lymphoid sub-population were included as supplementary Figure 1 for B cells and Supplementary Figure 2 for plasma cells.

Comment 3: Then exercise can increase the number of B cells expressing IgA in elderly mice, so whether sIgA in the intestine also increases? This result is very important to support the author’s conclusion.

Response 3: Thanks for your comment. Considering that IgA+ B cells are precursors of IgA+ plasma cells, we assumed that the IgA+ B cells remain more stable and they can differentiate into IgA producing plasma cells increasing the luminal IgA concentration. Together with the increase of IgA+ B cells %, elderly exercised mice expressed low levels of IL-4 mRNA versus sedentary elderly mice. Previously, it has been reported that IL-4 is a negative regulator of IgA plasma cell precursors [Suzuki K. et al., 2009]. Finally, in this study increased α-chain mRNA level found in exercised mice may result from the decrease of Pax5 mRNA favoring both the IgA plasma cell surviving and IgA CSR (Gommerman JL, et al., 2014). This information was included in the Discussion section, pag. 8, paragraph 3, lines 267-273.

  • Suzuki K, Fagarasan S. Diverse regulatory pathways for IgA synthesis in the gut. Mucosal Immunol (2009) 2:468–71. doi: 10.1038/MI.2009.107
  • Gommerman JL, Rojas OL, Fritz JH. Re-thinking the functions of IgA(+) plasma cells. Gut Microbes. 2014;5(5):652-62. doi: 10.4161/19490976.2014.969977. PMID: 25483334; PMCID: PMC4615259.

Comment 4. In Figure 2, 3 and 4, there was only the change of transcription level, and there was no experimental data of protein level change. In addition, are the changes in the transcription levels of IF-4 and IFN-γ related to Th cell response? The author should provide relevant explanations and proof.

Response 4. The reviewer is right. We are aware that one of the most important limitations of this study is that all factors associated to class switch recombination of IgM ® IgA, and factors associated to expression of pIgR were evaluated at transcriptional level. This limitation was included in the last paragraph of the Discussion section in pag. 9, paragraph 5, lines 338-341. Furthermore, in our study, levels of IL-4 and IFNg mRNA expression were analyzed as parameters associated with pIgR mRNA and not to T cell differentiation into Th2 or Treg phenotype.

Comment 5. (i) The important conclusion of the author 's article is that exercise leads to a change in the number of B cells expressing IgA in the intestine of aged mice, which down-regulates the intestinal pro-inflammatory response. (ii) However, in Figure 2 and Figure 4, the important pro-inflammatory factors in small intestine cells, IL-6 and TNF-α transcription levels did not change significantly. In addition, there was no data on the changes of pro-inflammatory factor IL-1. The authors need to provide relevant explanations and supplements for the changes of these important pro-inflammatory factors.

Response 5. Thank you for your comment. The reviewer is right regarding that, i) IgA has important anti-inflammatory effects through 1) immune exclusion via interacting with environmental molecules and pathogens; 2) anti-inflammation by sampling intestinal antigens to induce Th2 or regulatory T cell-biased mucosal immune responses; 3) homeostasis of commensals by enhancing the cross talk between the probiotic bacteria and the intestinal mucosa [Ren W, et al., 2016].

  1. ii) We know that the most important proinflammatory cytokines secreted during the aging are IL-6, TNFα, IFNg and IL-1êžµ, however the last cytokine was not evaluated because it does not play a role in the synthesis, secretion and transcytosis of IgA, which was the objective of this study. As it was included in the discussion, in the current contribution IL-6 mRNA expression at lamina propria was unchanged in exercised elderly mice, however in previous contributions moderate exercise increased IL-6 mRNA expression (but not TNFα and IFNg) in skeletal muscle, during young and aging stages [Petersen AM, et al., 2005]. Furthermore, exercise-induced IL-6 expression has anti-inflammatory effects favoring the secretion of IL-10, soluble TNF receptor and antagonist of IL-1 receptor [Petersen AM, et al., 2005]. See please pag. 8-9, 297-304. In this study MAE in elderly mice provided an unsignificant decrease of TNFα along with significant decrease IFNg suggesting a potential anti-inflammatory outcome in lamina propria of the small intestine.
  • Petersen AM, Pedersen BK. The anti-inflammatory effect of exercise. J Appl Physiol (1985). 2005 Apr;98(4):1154-62. doi: 10.1152/japplphysiol.00164.2004. PMID: 15772055.
  • Ren W, Wang K, Yin J, Chen S, Liu G, Tan B, et al. Glutamine-induced secretion of intestinal secretory immunoglobulin A: A mechanistic perspective. Front Immunol (2016) 7:503. doi: 10.3389/FIMMU.2016.00503

Comment 6. The author puts forward the conclusion that exercise leads to changes in the expression of IgA B cells in the intestinal tract of elderly rats, thus downregulating the intestinal pro-inflammatory response. Then, whether the changes of inflammatory factors (qPCR results) mentioned in this paper are related to the changes of IgA levels, the author’s data cannot fully support this view.

Response 6: Thank you for your comment. As it was commented in the response 4 and 5, the assessment of mRNA levels in some parameters as the main limitation of this study was included in discussion. See please pag. 9, paragraph 5, lines 338-341.

Comment 7. Does the sedentary mice mentioned in the author’s article do not do any treatment and are normally fed mice? If you can, please give a certain narrative in the article, do not cause ambiguity.

Response 7. Thank you for your comment. Groups of sedentary mice experimentally un-manipulated and fed with standard diet were included as control. See please pag. 10, lines 371-372. 

Reviewer 2 Report

Comments and Suggestions for Authors

Good morning to the authors,

I analyzed the manuscript with ID: ijms-3083506-peer-review-v1 antibiotics-3035974-peer-review-v1, entitled "Moderate Aerobic Exercise Induces the homeostatic IgA generation in Senile Mice" to be possibly published in the Journal International Journal of Molecular Sciences, Section: Molecular Biology; Special Issue - Aging: From Molecular Mechanisms, Pathophysiology to Novel Therapeutic Approaches,

I consider that:

1. The authors of the article proposed a much-discussed topic in today's medical scientific world: the factors that influence the development of the intestinal microbiome.

2. The article follows all the specific instructions of the journal presented in aims and scope, instructions for authors, and other information about the journal Int. J. Mol. Sci – MDPI.

3. Chapter 1. Introduction: The authors present relevant information for the chosen subject; in this article the influence of age and physical exercises on the increase of intestinal immunity.

4. Chapter 2. Results: The statistical analysis of the results is well presented and easy to understand in the text of the manuscript.

- In Figures 1 (a-f), Figure 2 (a-h), and Table 1 the results are correctly presented and are suggestive of the conducted study. They are very well presented.

5. Chapter 3. Discussions: The authors exemplified and compared all the results obtained by them with the studies and assessments of other authors, according to the bibliography.

6. Chapter 4. Materials and Methods: The data presented in this chapter are very well structured and coherent.

- The study was carried out in dynamics (24 months), on experimental animals: six-week-old male BALB/c mice.

- All norms/rules for the protection of laboratory animals regarding feeding, rearing, and physical exercises were respected.

- The authors used in their study a sufficiently large number of mice (n=40) divided into several subgroups (5), purchased from an accredited Institute.

- They used standardized and medically validated working protocols regarding the breeding/development of laboratory animals (mice).

- The authors used standardized work protocols regarding the collection of biological samples, protocols that have been scientifically validated by other authors.

- The working protocols for the processing of the taken biological samples comply with all the specific standards. The medical laboratory equipment used is of high performance.

- The kits, culture media, and reagent kits are supplied by specialized and prestigious companies.

7. The bibliography chosen by the authors corresponds to the requirements and refers to the subject of the article.

8. All authors have made a fair contribution to the study.

9. The authors obtained the approval of the Ethics Committee of their Institute for this study.

10. The authors also received funding for this study demonstrating professionalism and medical scientific relevance.

In conclusion:

ACCEPT for possible publication of the Journal Int. J. Mol. Sci for this article.

Congratulations to all authors for this article!

Felicitaciones a todos los autores por este artículo!

Author Response

Comment 1: I analyzed the manuscript with ID: ijms-3083506-peer-review-v1 antibiotics-3035974-peer-review-v1, entitled "Moderate Aerobic Exercise Induces the homeostatic IgA generation in Senile Mice" to be possibly published in the Journal International Journal of Molecular Sciences, Section: Molecular Biology; Special Issue - Aging: From Molecular Mechanisms, Pathophysiology to Novel Therapeutic Approaches,

I consider that:

  1. The authors of the article proposed a much-discussed topic in today's medical scientific world: the factors that influence the development of the intestinal microbiome.

  1. The article follows all the specific instructions of the journal presented in aims and scope, instructions for authors, and other information about the journal Int. J. Mol. Sci – MDPI.

  1. Chapter 1. Introduction: The authors present relevant information for the chosen subject; in this article the influence of age and physical exercises on the increase of intestinal immunity.

  1. Chapter 2. Results: The statistical analysis of the results is well presented and easy to understand in the text of the manuscript.
  • In Figures 1 (a-f), Figure 2 (a-h), and Table 1 the results are correctly presented and are suggestive of the conducted study. They are very well presented.

  1. Chapter 3. Discussions: The authors exemplified and compared all the results obtained by them with the studies and assessments of other authors, according to the bibliography.

  1. Chapter 4. Materials and Methods: The data presented in this chapter are very well structured and coherent.
  • The study was carried out in dynamics (24 months), on experimental animals: six-week-old male BALB/c mice.
  • All norms/rules for the protection of laboratory animals regarding feeding, rearing, and physical exercises were respected.
  • The authors used in their study a sufficiently large number of mice (n=40) divided into several subgroups (5), purchased from an accredited Institute.
  • They used standardized and medically validated working protocols regarding the breeding/development of laboratory animals (mice).
  • The authors used standardized work protocols regarding the collection of biological samples, protocols that have been scientifically validated by other authors.
  • The working protocols for the processing of the taken biological samples comply with all the specific standards. The medical laboratory equipment used is of high performance.
  • The kits, culture media, and reagent kits are supplied by specialized and prestigious companies.

  1. The bibliography chosen by the authors corresponds to the requirements and refers to the subject of the article.

  1. All authors have made a fair contribution to the study.

  1. The authors obtained the approval of the Ethics Committee of their Institute for this study.

  1. The authors also received funding for this study demonstrating professionalism and medical scientific relevance.

In conclusion:

ACCEPT for possible publication of the Journal Int. J. Mol. Sci for this article.

Congratulations to all authors for this article!

Felicitaciones a todos los autores por este artículo!

Response 1: Thank you very much for your thorough revision of this manuscript. We appreciated so much your fine comments.

Reviewer 3 Report

Comments and Suggestions for Authors

The authors of this paper wanted to see how long-term MAE affected TI-IgA production in young (3 months old) BALB/c mice that were exercised until adulthood or aging. Flow cytometry was used to identify sub-populations of B and plasma cells in the small intestine, as well as molecular factors related to class switch recombination and IgA synthesis. Epithelial cells were used to evaluate IgA-transitosis using RT-qPCR. Under senescence conditions, MAE promoted B cells and IgA+B cells, as well as APRIL, which may improve the intestinal response and reduce inflammation, presumably through pIgR downmodulation. So the authors concluded that, MAE can improve IgA levels and reduce pro-inflammatory cytokine expression, promoting homeostasis during aging.

The study is generally well-written, interesting to the scientific community, and deserves to be published after minor revisions. 

Comments on the Quality of English Language

minor corrections required

Author Response

Comment 1: The study is generally well-written, interesting to the scientific community, and deserves to be published after minor revisions.

Response 1: Thank you very much for your minacious revision of this manuscript. We appreciated so much your fine comments.

Comment 2: Comments on the Quality of English Language minor corrections required.

Response 2: Thank you for your comments. Quality of English Language was revised.

Reviewer 4 Report

Comments and Suggestions for Authors

This is an interesting study on the role of moderate aerobic exercise and IgA T-independent pathway on mucosal homeostatic balance in young, adult and senile mice. The paper is well written with some minor mistakes (parenthesis not closed, abbreviations not stated,  tense changes and some poorly written words) which should be corrected through the text. Authors concluded that moderate aerobic exercise may improve the T-cell indipendent IgA-response and promoting the luminal IgA production and suppressing pro-inflammatory cytokines expression, ultimately favoring a homeostatic environment in aging animal models. 

Some minor revisions and concerns:

Introduction: IgA synthesis and cell-interactions are complex (as highlighted in the introduction), this section could benefit from a figure that clarifies the main concepts explained in here . Furthermore, there are some interesting old and new studies on this topic which have not been taken into consideration here, especially some of those about immunosenescence and mucosal immunity.  

A point of concern is about the diet of the animal subjects, have you standardized the nutrients given to the mice? Nutrition has a notable impact on both health and senescence.

Where should future research be directed? How can this research impact human health? Please add your thoughts in the conclusion paragraph. 

Comments on the Quality of English Language

The text is clear, there are some minor mistakes that should be corrected.

Author Response

Comment 1: This is an interesting study on the role of moderate aerobic exercise and IgA T-independent pathway on mucosal homeostatic balance in young, adult and senile mice. The paper is well written with some minor mistakes (parenthesis not closed, abbreviations not stated, tense changes and some poorly written words) which should be corrected through the text. Authors concluded that moderate aerobic exercise may improve the T-cell independent IgA-response and promoting the luminal IgA production and suppressing pro-inflammatory cytokines expression, ultimately favoring a homeostatic environment in aging animal models.

Response 1: Thank you very much for your minacious revision of this manuscript. We appreciated so much your fine comments.

Some minor revisions and concerns:

Comment 2: Introduction: i) IgA synthesis and cell-interactions are complex (as highlighted in the introduction), this section could benefit from a figure that clarifies the main concepts explained in here. ii) Furthermore, there are some interesting old and new studies on this topic which have not been taken into consideration here, especially some of those about immunosenescence and mucosal immunity. 

Response 2: Thank you for your comments.  i) we try to represent this information in the graphical abstract and include some references where representative figures are included. ii) As far as we know, we made a rigorous selection of old and new manuscripts related to the topic of our study focused particularly with exercise and aging and IgA which were included in the Introduction and Discussion sections.

Comment 3: A point of concern is about the diet of the animal subjects, have you standardized the nutrients given to the mice? Nutrition has a notable impact on both health and senescence.

Response 3. Thank you for your comment. As the reviewer has commented, diet has a substantive effect that can modify the response to exercise and aging. In this study, mice fed during their complete life cycle standard diet formulated by Laboratory Rodent Diet 5001, recommended to assure minimal inherent biological variation in long-term studies (https://www.labdiet.com/product/detail/5001-laboratory-rodent-diet). This information was provided in sub-section 4.1 Animal of the Materials and Methods section in the first version of the study.

Comment 4: Where should future research be directed? How can this research impact human health? Please add your thoughts in the conclusion paragraph.

Response 4:  Thak you for your comment. In the conclusion section, the next information was included in the revised version: “Exercise induced-homeostatic IgA production during aging could be a protective mechanism against pathogens and inflammatory diseases in elderly people”. See please pag. 12, lines 449-451

Comment 4: Comments on the Quality of English Language

The text is clear, there are some minor mistakes that should be corrected.

Response 4:  Thank you for your comment. Quality of English Language was minacious revised.

Round 2

Reviewer 1 Report

Comments and Suggestions for Authors The authors have solved all my comments。